# EpiLoop: A Diagnostic Protocol for Measuring Epistemic Lock-in in Iterative Search Agents

## Abstract

Iterative search agents repeatedly retrieve evidence, update context, and decide whether to keep searching or to answer. Existing evaluations emphasize final-answer accuracy or retrieval quality, but the agent's evolving epistemic state remains implicit. We study *epistemic lock-in*: a reliability failure in which an early hypothesis shapes later queries, evidence selection, confidence estimates, and stopping decisions, so that additional retrieval reinforces rather than revises the answer. We introduce a diagnostic protocol that measures lock-in through observable trajectory proxies, with Hypothesis Change Rate (HCR) as the primary measure of hypothesis instability. We also present EpiLoop, a lightweight inference-time actionability probe that tracks hypothesis, confidence, source coverage, closure judgment, and refutation intent. In a controlled HoVer-derived study, EpiLoop matches equal-budget forced extra search in final accuracy (75.0% vs. 75.0%, paired bootstrap 95% CI [-0.100, +0.100]) while producing lower HCR (0.083 vs. 0.121). Cross-model checks and external validation slices on FEVER, SciFact, and VitaminC show the same qualitative pattern: HCR decreases, while accuracy effects remain dataset- and parser-dependent. These results do not show that EpiLoop is more accurate than extra search. Rather, they show that explicit epistemic-state tracking exposes and reduces a trajectory-level instability that final-answer accuracy alone misses.

## 1 Introduction

Retrieval-augmented language-model agents increasingly operate as iterative systems. They plan, issue a query, inspect retrieved evidence, update an answer, and decide whether to continue searching. This design appears to give the model repeated chances to correct early mistakes. Yet the same loop can create path dependence: once a plausible hypothesis appears early, later queries may narrow around it, retrieved evidence may become redundant, confidence may rise for the wrong reason, and stopping decisions may reflect accumulated familiarity rather than sufficient evidence.

We call this failure mode *epistemic lock-in.* Lock-in is not a single retrieval miss, a hallucination, static confirmation bias, or premature stopping in isolation. It is a feedback pathology. The agent's current belief state shapes future retrieval, and future retrieval further shapes the belief state. In an iterative search agent, reliability therefore depends not only on whether relevant documents are available, but also on whether the loop supports effective hypothesis revision.

The concept draws structural parallels with long-studied phenomena in human cognition. Confirmation bias is the tendency to seek, interpret, and remember information that confirms a prior belief (Nickerson, 1998; Kahneman, 2011). Belief perseverance describes how initial beliefs can survive exposure to disconfirming evidence (Anderson et al., 1980). We do not claim that language-model agents have human beliefs. Instead, these literatures motivate a computational diagnostic question: when an agent's own provisional answer influences query generation, evidence use, confidence, and closure, does the loop display a measurable analogue of belief persistence?

Most current evaluations of retrieval-augmented generation and search agents emphasize final-answer accuracy, retrieval recall, citation quality, or aggregate task success (Lewis et al., 2020; Trivedi et al., 2022; Yao

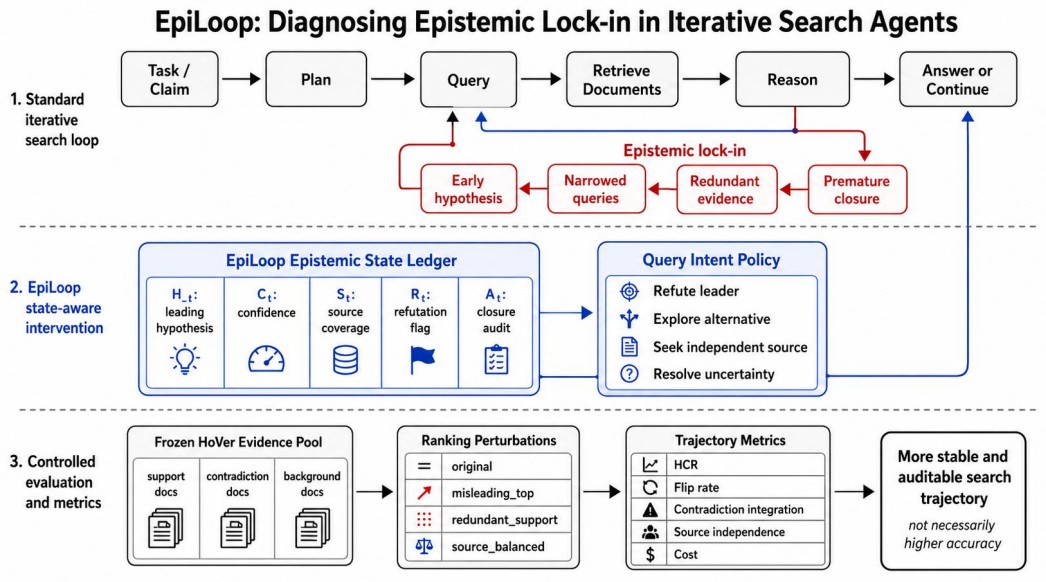

Figure 1: EpiLoop instruments an iterative search loop with five epistemic-state trackers: hypothesis, confidence, source coverage, closure judgment, and refutation intent. The controller does not replace the model or retriever. It uses the tracked state to choose the next search intent and to audit stopping decisions.

et al., 2022; Asai et al., 2023; Jiang et al., 2023). These measures are necessary but incomplete. Two agents can achieve the same accuracy while following very different epistemic trajectories. One may revise after encountering counter-evidence. Another may lock onto an early answer, keep retrieving near-duplicates, and stop with high confidence. The latter behavior is risky in autonomous settings because it wastes compute, obscures uncertainty, and makes answers sensitive to document ordering or backend ranking.

This paper contributes a diagnostic framing rather than a new search-agent architecture. We treat epistemic state as latent and therefore not directly observable. The protocol instead measures behavioral proxies from the trajectory: leading hypotheses, hypothesis changes, source groups, contradiction exposure, confidence changes, and stopping behavior. The primary stability metric is **Hypothesis Change Rate (HCR)**, the proportion of adjacent trajectory steps in which the leading hypothesis changes. We interpret HCR alongside ranking sensitivity, contradiction integration, source independence, closure behavior, and cost.

To test whether the diagnosed state is actionable, we introduce EpiLoop. EpiLoop is not a new model, retriever, benchmark, or fine-tuning method. It is a lightweight inference-time actionability probe: a controller that tracks observable epistemic state and uses it to choose the next search intent. Its purpose is empirical. If lightweight state tracking changes the trajectory relative to standard iteration and equal-budget extra search, then epistemic lock-in is not merely a post-hoc description; it is an actionable reliability signal.

Our contributions are:

- **Epistemic lock-in definition.** We formalize epistemic lock-in as a path-dependent reliability failure in iterative search agents. An early hypothesis shapes subsequent queries, evidence selection, confidence, and stopping decisions, causing additional retrieval to reinforce rather than revise the answer.

- **Diagnostic protocol and metrics.** We propose a controlled perturbation protocol in which the evidence pool is fixed while ranking and contradiction timing are manipulated, and we define trajectory-level metrics for hypothesis instability, ranking sensitivity, contradiction integration, source independence, and closure behavior.

- **EpiLoop actionability probe.** We design a lightweight inference-time controller that tracks five epistemic-state variables and uses rule-based search intents to guide retrieval and stopping decisions.[1]

- **Empirical findings.** In a 40-task HoVer-derived study, EpiLoop matches equal-budget forced extra search in final accuracy while reducing HCR from 0.121 to 0.083. Cross-model validation and additional FEVER/SciFact/VitaminC checks show consistent HCR reduction, suggesting that the stability effect is not a single-model or single-dataset artifact. Accuracy effects remain dataset- and parser-dependent.

Our central claim is deliberately narrow. EpiLoop does not significantly improve final-answer accuracy over equal-budget forced extra search in the current study. Its value is diagnostic: it makes a hidden trajectory property measurable and shows that explicit epistemic-state tracking can reduce hypothesis instability under controlled budgets and across several external validation settings.

The remainder of the paper proceeds as follows. Section 2 defines epistemic lock-in and the failure modes the diagnostic separates. Section 3 introduces the trajectory metrics. Section 4 describes EpiLoop as a lightweight actionability probe. Section 5 gives the experimental setup, and Section 6 reports controlled, cross-model, and external-validation results.

## 2 Epistemic Lock-in

### 2.1 Iterative Search as Epistemic Process Control

An iterative search agent can be represented as a loop over rounds. At each round $t$, the agent maintains a behavioral state

$$s_t = (h_t, c_t, E_t, G_t),$$

where $h_t$ is the leading hypothesis, $c_t$ is self-reported confidence, $E_t$ is collected evidence, and $G_t$ is the set of source groups observed. The agent issues a query $q_{t+1} = \pi_q(s_t)$, receives documents $d_{t+1} = \text{RETRIEVE}(q_{t+1})$, and updates its state:

$$s_{t+1} = \pi_u(s_t, d_{t+1}).$$

The loop terminates when a stopping condition $f_{\text{stop}}(s_t)$ is met. Standard implementations often treat $s_t$ as accumulated text. EpiLoop instead treats the loop as an epistemic process in which the agent maintains, revises, and closes over hypotheses.

More context is not equivalent to better epistemic state. Additional documents can add independent evidence, or they can repeat the same source group. They can challenge the current hypothesis, or they can reinforce it. They can justify higher confidence, or they can inflate confidence through redundancy. A reliability-oriented evaluation should therefore ask whether each search step increases the agent's justified ability to revise or close the answer.

### 2.2 Definition and Observable Symptoms

We define epistemic lock-in as a *theoretical model* of path-dependent reliability failure: an early hypothesis becomes increasingly resistant to revision because subsequent query generation, evidence selection, confidence estimation, and stopping decisions are conditioned on the preceding search trajectory. This model has four properties.

First, lock-in is dynamic. It unfolds over multiple rounds rather than appearing as a single wrong answer. Second, it is path-dependent. The same evidence set can produce different final states if the agent's query sequence evolves differently. Third, it is epistemic rather than purely retrieval-based: the relevant question is not only whether contradiction evidence is retrieved, but whether it is integrated into the agent's answer or confidence. Fourth, it is latent. Because we cannot directly observe a model's internal belief state, lock-in must be diagnosed through behavioral proxies.

---

[1]Code: https://anonymous.4open.science/r/EpiLoop-B443.

| Property | Position bias | Epistemic lock-in |
|---|---|---|
| Timescale | Single context or ranked list | Multi-round search trajectory |
| Mechanism | Attention or salience by position | Hypothesis-conditioned retrieval and closure |
| Observable unit | Output sensitivity to item order | Query, evidence, confidence, and stopping path |
| Key risk | Overweighting early or central evidence | Reinforcing an early hypothesis through the loop |
| Diagnostic need | Reorder or rebalance context | Audit revision, source coverage, contradiction, closure |

Table 1: Position bias can initiate lock-in, but lock-in adds a feedback mechanism across search rounds. The same early ranking effect becomes more consequential when it shapes future queries and stopping decisions.

To make the model empirically testable, we operationalize a ranking-sensitivity condition. Let $\tau = (s_1, q_2, d_2, s_2, \ldots, s_T)$ be a trajectory, $\mathcal{P}$ be a fixed evidence pool, and $\mathcal{D}_a, \mathcal{D}_b \in \text{Perm}(\mathcal{P})$ be two document orderings. If

$$h_T(\mathcal{D}_a) \neq h_T(\mathcal{D}_b),$$

then the terminal hypothesis is sensitive to document order under fixed evidence availability. This condition is not sufficient to prove lock-in by itself. It can also reflect stochasticity, ambiguous evidence, or brittle prompting. The full theoretical model is therefore evaluated through a constellation of trajectory symptoms: hypothesis instability or collapse, negative-evidence blindness, evidence saturation, backend sensitivity, and premature closure.

### 2.3 Failure Modes the Diagnostic Separates

The diagnostic protocol is designed to distinguish several failures that are often conflated.

**Retrieval absence** occurs when the needed evidence is not available in the local context. Lock-in cannot be diagnosed from missed evidence alone because the agent never had an opportunity to revise. **Contradiction retrieval without integration** occurs when counter-evidence is present but the agent does not acknowledge or use it. This is closer to lock-in because the opportunity for revision exists. **Redundant support** occurs when many retrieved snippets support the same surface claim but come from the same source group or repeat the same fact. **Premature closure** occurs when the agent stops before observing independent evidence or resolving a contradiction. EpiLoop targets these latter three failures by making source coverage, contradiction status, and closure justification explicit.

### 2.4 Relation to Position Bias

Epistemic lock-in can include position effects, but it is not reducible to single-step position bias. Position bias concerns how a model weights information depending on where it appears in a context window or ranked list, as in long-context position effects and "lost in the middle" behavior (Liu et al., 2023). Lock-in is a multi-round control pathology: an early position-induced hypothesis can change later queries, which then changes the future evidence distribution, confidence trajectory, and stopping decision. Thus, position bias can be one trigger for lock-in, but the diagnostic target is the feedback loop that follows.

## 3 Diagnostic Protocol and Metrics

### 3.1 Controlled Evidence Protocol

The diagnostic protocol fixes the local evidence pool and perturbs the search trajectory. In our HoVer-derived setup (Jiang et al., 2020), each task contains a claim, a support/refute label, documents with source-group metadata, and stance annotations. The same documents are then presented under four rankings: *original*, *misleading-top*, *redundant-support*, and *source-balanced*. Because the content set is fixed, differences across conditions can be attributed to exposure order and loop dynamics rather than document availability.

This design intentionally trades ecological realism for causal clarity. Live web search introduces confounds from index drift, snippet generation, search-engine ranking, recency bias, and page availability. A frozen local corpus cannot evaluate those phenomena, but it can isolate the loop-level question: when the same evidence exists, does the agent's trajectory make revision more or less likely?

### 3.2 Hypothesis Change Rate

Let $h_t$ be the leading hypothesis at trajectory step $t$, where the label space is $h_{\text{support}}$, $h_{\text{refute}}$, or $h_{\text{uncertain}}$. For a trajectory of $T$ hypothesis-bearing steps, we define:

$$\text{HCR} = \frac{1}{T-1} \sum_{t=1}^{T-1} \mathbf{1}[h_t \neq h_{t+1}].$$

HCR is a trajectory-level proxy for hypothesis instability. High HCR indicates frequent changes in the leading hypothesis; low HCR indicates a stable trajectory. Low HCR is not automatically good. It can reflect justified convergence or unjustified lock-in. High HCR is not automatically bad. It can reflect healthy evidence-responsive revision or unstable oscillation. We therefore interpret HCR alongside contradiction exposure, source independence, closure behavior, and final accuracy.

### 3.3 Additional Metrics

**Ranking sensitivity** measures whether the final leading hypothesis changes when document order changes under a fixed evidence set. Let $h_T(\mathcal{D})$ denote the terminal hypothesis given document ordering $\mathcal{D}$. The flip rate between orderings $\mathcal{D}_a$ and $\mathcal{D}_b$ is:

$$\text{FlipRate}(\mathcal{D}_a, \mathcal{D}_b) = \frac{1}{N} \sum_{i=1}^{N} \mathbf{1}[h_T^{(i)}(\mathcal{D}_a) \neq h_T^{(i)}(\mathcal{D}_b)].$$

**Contradiction integration** distinguishes seeing counter-evidence from using it: whether the trajectory explicitly acknowledges, qualifies, or revises in response to evidence that weakens the leading hypothesis. **Source coverage** captures how many distinct source groups have been observed. **Source independence** is the normalized form:

$$\text{SrcIndep} = \frac{|\text{unique source groups retrieved}|}{|\text{retrieved documents}|}.$$

**Closure** asks whether stopping is justified by source coverage, contradiction integration, source independence, and unresolved uncertainties. **Cost** reports search rounds, retrieval calls, latency, and token use.

### 3.4 Worked Interpretation

Consider two trajectories with the same final answer. In the first, the agent starts uncertain, retrieves evidence from two independent source groups, encounters a contradictory snippet, revises its hypothesis, lowers confidence, then finds resolving evidence and stops. This trajectory may have moderate HCR, but the changes are evidence-responsive. In the second, the agent starts with a support hypothesis, retrieves three snippets from the same page family, ignores a contradictory sentence, and stops with high confidence. This trajectory may have low HCR, but low HCR here is consistent with lock-in rather than reliability. The protocol treats HCR as a signpost, not a verdict.

## 4 EpiLoop Actionability Probe

EpiLoop is a lightweight inference-time wrapper around an iterative search agent. It does not replace the underlying LLM, retriever, or reasoning prompt. Instead, it tests whether observable epistemic state can guide the loop in ways that reduce lock-in symptoms.

The controller tracks five state variables: the current leading hypothesis, self-reported confidence, source coverage, closure judgment, and refutation intent. These fields are extracted from structured model outputs and trajectory metadata. They are behavioral approximations, not privileged access to internal model state.

At each round, EpiLoop selects a search intent rather than directly optimizing a learned policy. Possible intents include seeding hypotheses, supporting the current leader, refuting the leader, exploring an alternative, resolving uncertainty, seeking independent sources, and checking source dependence. The rule-based policy is intentionally simple: if no hypothesis exists, seed hypotheses; if the answer appears ready, run a closure audit; if lock-in risk is high, refute the leader; if evidence is redundant, seek independent sources.

### 4.1 Mechanisms

The **confidence ceiling** prevents premature closure when confidence is high but source coverage is thin:

$$\text{if } c_t > \tau_{\text{conf}} \wedge |G_t| \leq \tau_{\text{source}}$$
$$\rightarrow \textsc{SeekIndependentSources}().$$

The **closure audit** checks whether termination is justified:

$$f_{\text{stop}}(s_t) = (c_t > \tau_{\text{close}}) \wedge (|G_t| \geq 2) \wedge \neg \textsc{UnresolvedContradiction}(s_t).$$

The agent must have sufficient confidence, evidence from multiple independent source groups, and no unresolved contradiction.

The **refutation intent** triggers when the leading hypothesis has persisted for $\kappa$ consecutive steps while contradictory evidence has appeared:

$$\text{if } \Delta_{\text{stable}}(h_t) \geq \kappa \wedge \textsc{HasContradiction}(E_t)$$
$$\rightarrow q_{t+1} = \textsc{RefuteQuery}(h_t).$$

The thresholds are fixed before evaluation and held constant across experiments. We do not claim that they are optimal; they serve to test whether a simple state-aware probe can change the trajectory under a matched retrieval budget.

---

**Algorithm 1** EpiLoop actionability probe

**Require:** claim $x$, evidence pool $\mathcal{P}$, base agent $\mathcal{A}$, maximum rounds $T$
 1: $s_0 \leftarrow$ initialize epistemic state for $x$
 2: **for** $t = 0$ **to** $T - 1$ **do**
 3:     intent $\leftarrow$ choose_intent($s_t$)
 4:     $q_{t+1} \leftarrow \mathcal{A}.\textsc{GenerateQuery}(x, s_t, \text{intent})$
 5:     $D_{t+1} \leftarrow \textsc{Retrieve}(q_{t+1}, \mathcal{P})$
 6:     $s_{t+1} \leftarrow \mathcal{A}.\textsc{UpdateState}(x, s_t, D_{t+1})$
 7:     **if** $\textsc{ClosureAudit}(s_{t+1})$ is pass **then**
 8:         **return** final answer from $s_{t+1}$
 9:     **end if**
10: **end for**
11: **return** final answer from $s_T$

---

## 5 Experimental Setup

### 5.1 Dataset and Task Construction

We evaluate on a 40-task sample derived from HoVer. The task slice contains 141 documents with corrected source-group metadata derived from Wikipedia page titles. Each task is converted into a local frozen-corpus

| Dataset | Quality profile | Role in paper |
|---|---|---|
| HoVer (Jiang et al., 2020) | Corrected source groups; controlled ranking perturbations; explicit stance labels. | Primary diagnostic experiment. |
| FEVER (Thorne et al., 2018) | Mixed-stance fact-verification items; one negation item in the validation slice. | Directional external check. |
| SciFact (Wadden et al., 2020) | No mixed stances or oracle-document mismatch in the validation slice; one negation item. | Cleanest external validation. |
| VitaminC (Schuster et al., 2021) | Systematic mixed stances; systematic claim-variant mismatch; oracle labels are heavily skewed toward supported. | Conflict-heavy stress test, not an accuracy benchmark. |

Table 2: Dataset roles and quality profile. VitaminC is retained for trajectory stress testing, but its strict accuracy is treated as noisy because internally conflicting evidence can make oracle-supported examples appear locally unsupported.

search problem. The HoVer oracle label is mapped to $h_{\text{support}}$ or $h_{\text{refute}}$; the trajectory schema also includes $h_{\text{uncertain}}$ for cases where the agent cannot justify either label from observed evidence.

The task construction fixes two earlier sources of experimental ambiguity. First, source groups are real Wikipedia page titles rather than synthetic document identifiers, making source-independence measurements meaningful. Second, stance labels distinguish contradiction retrieval from contradiction integration. A trajectory can retrieve a contradictory document and still fail to integrate it into the answer.

**External validation slices.** For external validation, we additionally run the primary full-loop comparison on FEVER (Thorne et al., 2018), SciFact (Wadden et al., 2020), and VitaminC (Schuster et al., 2021) using DeepSeek V4 Pro, the misleading-top ranking, max-rounds 8, and temperature 0. These datasets are used as transportability checks rather than replacements for the HoVer controlled experiment. We therefore report them as external validation slices rather than benchmark-scale accuracy evaluations.

## 5.2 Answer Scoring and VitaminC Ambiguity

**Strict oracle scoring.** For strict accuracy, we retain each dataset's official oracle label. If an item has oracle label SUPPORTED and the model answers "not supported," "evidence contradictory," or "insufficient evidence," the prediction is counted as an oracle disagreement. We do not relabel these outputs as correct. This rule preserves benchmark comparability and prevents post-hoc adjudication.

**VitaminC interpretation.** VitaminC requires a separate interpretation layer. In this validation slice, VitaminC items systematically contain mixed support/refute stances and contrastive claim-variant mismatch, while the oracle distribution is overwhelmingly SUPPORTED. Thus, an answer such as "the evidence is contradictory" may reflect a reasonable response to the local evidence bundle even though it disagrees with the official oracle. We therefore report VitaminC strict accuracy only as a secondary, noisy outcome and do not use it to argue that one agent is more correct than another. The primary VitaminC evidence is trajectory-level: whether EpiLoop reduces hypothesis churn under conflicting evidence.

## 5.3 Agents and Conditions

We compare closed-book answering, single-round retrieval, no-history iteration, standard iteration, cautious prompting, diverse-query prompting, forced extra search, and EpiLoop. The main intervention comparison uses equal maximum rounds of 8 under the *misleading-top* ranking. EpiLoop and forced extra search have matched realized retrieval calls in the primary DeepSeek comparison (3.0 average retrieval calls each).

The conditions serve different diagnostic purposes. Closed-book answering estimates the no-retrieval floor. Single-round retrieval tests whether one exposure to evidence is enough. No-history iteration tests whether repeated retrieval without accumulated context avoids path dependence. Standard iteration represents ordinary context accumulation. Forced extra search controls for the possibility that EpiLoop's benefit comes

| Block | Purpose | Main variables | Baselines |
|-------|---------|----------------|-----------|
| RQ1 | loop effect | mode $\times$ ranking | closed-book |
| RQ2 | order sensitivity | ranking $\times$ contradiction timing | early/original |
| RQ3 | intervention | agent under equal max rounds | standard, forced-extra |
| RQ4 | mechanism check | EpiLoop ablations | full EpiLoop |
| RQ5 | model check | model family $\times$ agent | standard, forced-extra |
| RQ6 | dataset check | dataset $\times$ agent | standard, forced-extra |

Table 3: Experimental design matrix. RQ1–RQ4 use DeepSeek V4 Pro and the corrected 40-task HoVer-derived slice; RQ5 repeats the primary intervention comparison across three model families; RQ6 repeats it on FEVER, SciFact, and VitaminC. The study is designed as a diagnostic evaluation rather than a benchmark-scale accuracy comparison.

only from spending more retrieval budget. Cautious and diverse-query prompting test whether generic uncertainty or diversity instructions replicate the effect of state-aware tracking.

### 5.4 Research Questions and Statistics

The experiments address six questions:

RQ1 How do closed-book, single-round, no-history, and full-loop search differ in accuracy and trajectory behavior?

RQ2 Does document ranking change final hypotheses when the evidence set is fixed?

RQ3 Does EpiLoop differ from standard iteration and equal-budget forced extra search?

RQ4 Which EpiLoop mechanisms matter in the pilot ablation?

RQ5 Does HCR reduction persist across model families?

RQ6 Does HCR reduction transport to additional fact-verification datasets?

For the DeepSeek study, we report bootstrap 95% confidence intervals with 10,000 resamples and seed 42. For paired accuracy comparisons, we use paired bootstrap difference intervals and McNemar tests. We report Cohen's $h$ for accuracy differences and Cohen's $d$ for HCR differences. Cross-model accuracy estimates are exploratory because automatic oracle matching covers only 15–21 valid outputs per model-agent condition. In the external validation runs, accuracy is computed by a deterministic final-answer classifier; HCR is computed directly from logged hypothesis trajectories.

### 5.5 Power and Scope

At $N = 40$ and $\alpha = 0.05$ (two-sided McNemar), the study has 80% power to detect an accuracy difference of approximately $\pm 27.5$ percentage points, which is a large effect. Smaller effects fall below the detectable threshold at this sample size. For HCR comparisons, the minimum detectable effect size is approximately $d \approx 0.45$ under a paired normal approximation. The observed HCR differences are therefore near the study's detection boundary. We report these estimates so that non-significant accuracy results are not over-interpreted as evidence of equivalence.

## 6 Results

### 6.1 Retrieval Helps, but Does Not Fully Explain Lock-in

The loop-mode comparison shows that retrieval access matters. Closed-book answering reaches 52.5% accuracy, while full-loop retrieval reaches 75.0% on the original ranking. The paired difference between full-loop

| Condition | Acc. | 95% CI | HCR | Steps | Ret. |
|---|---|---|---|---|---|
| closed-book | 52.5 | [37.5, 67.5] | 0.000 | 2.0 | 0.0 |
| single-round, original | 67.5 | [52.5, 82.5] | 0.425 | 4.0 | 1.0 |
| no-history, original | 65.0 | [50.0, 80.0] | 0.147 | 5.4 | 1.2 |
| full-loop, original | 75.0 | [60.0, 87.5] | 0.130 | 5.5 | 1.3 |

Table 4: RQ1 loop-mode comparison on the 40-task HoVer-derived slice. Accuracy is reported as a percentage; HCR is lower-is-more-stable; Steps and Ret. are averages. Retrieval substantially improves accuracy over closed-book answering, but loop dynamics still require trajectory-level analysis.

| Ranking | Timing | Flip | Acc. | 95% CI |
|---|---|---|---|---|
| original | early | 0.0 | 77.5 | [65.0, 90.0] |
| original | late | 12.5 | 75.0 | [60.0, 87.5] |
| original | present-not-top | 20.0 | 72.5 | [57.5, 85.0] |
| misleading-top | early | 10.0 | 75.0 | [60.0, 87.5] |
| misleading-top | late | 15.0 | 72.5 | [57.5, 85.0] |
| misleading-top | present-not-top | 12.5 | 75.0 | [60.0, 87.5] |
| redundant-support | early | 17.5 | 75.0 | [60.0, 87.5] |
| redundant-support | late | 10.0 | 75.0 | [60.0, 87.5] |
| redundant-support | present-not-top | 20.0 | 72.5 | [57.5, 85.0] |
| source-balanced | early | 15.0 | 70.0 | [55.0, 85.0] |
| source-balanced | late | 7.5 | 72.5 | [57.5, 85.0] |
| source-balanced | present-not-top | 12.5 | 77.5 | [65.0, 90.0] |

Table 5: RQ2 ranking sensitivity under fixed evidence. Flip is the final-hypothesis flip rate relative to the early/original condition. The non-monotonic pattern across timing conditions suggests that ranking effects are mediated by trajectory dynamics, not just by whether contradiction appears early or late.

and closed-book is +22.5 percentage points with bootstrap $p = 0.0094$ and McNemar $p = 0.0265$. However, retrieval access alone does not resolve the trajectory problem. Document order still changes final hypotheses, and different agents with similar accuracy can have substantially different HCR.

## 6.2 Fixed Evidence, Different Ranking, Different Hypotheses

Under the standard full-loop agent, changing document order changes final hypotheses for a non-trivial fraction of tasks. Misleading-top rankings produce 10.0%–15.0% flip rates depending on contradiction timing; redundant-support and source-balanced rankings produce early-condition flip rates of 17.5% and 15.0%, respectively. The effect is not monotonic across contradiction timing, suggesting that lock-in is not simply "contradiction appears too late." Document order interacts with the agent's evolving commitment state.

## 6.3 EpiLoop Reduces HCR Without Improving Accuracy Over Extra Search

**Accuracy is not the win condition.** The central result is a stability-accuracy separation. The paired accuracy comparison between EpiLoop and forced extra search is exactly null in this pilot: $\Delta = +0.000$, 95% CI [-0.100, +0.100], bootstrap $p = 1.0000$, McNemar $p = 0.6171$, Cohen's $h = 0.0000$. This should not be hidden. State-aware control is not more accurate than simply spending the same retrieval budget.

**Trajectory stability is the signal.** EpiLoop has the lowest HCR among all intervention conditions. The HCR effect size is medium against the standard agent ($d = 0.5471$) and small against forced extra search ($d = 0.3805$). It also has the lowest hypothesis deviation from the standard agent, with a 10.0% flip rate compared with 15.0% for forced extra search, 20.5% for diverse-query prompting, and 33.3% for cautious prompting.

**Conditioning on correctness.** Table 7 addresses the main construct-validity ambiguity. EpiLoop's lower HCR is not purely a correctness signal: wrong-and-stable trajectories occur in the EpiLoop condition as well

| Agent | Acc. | 95% CI | HCR | Flip | Steps | Ret. |
|---|---|---|---|---|---|---|
| standard | 70.0 | [55.0, 85.0] | 0.151 | – | 5.5 | 1.3 |
| cautious | 74.4 | [59.0, 87.2] | 0.221 | 33.3 | 6.6 | 1.9 |
| diverse-query | 74.4 | [61.5, 87.2] | 0.146 | 20.5 | 6.9 | 2.0 |
| forced-extra | 75.0 | [60.0, 87.5] | 0.121 | 15.0 | 8.0 | 3.0 |
| **EpiLoop** | **75.0** | [60.0, 87.5] | **0.083** | **10.0** | 8.0 | 3.0 |

Table 6: Primary intervention comparison on the misleading-top ranking. Accuracy and flip rate are percentages; HCR is lower-is-more-stable; Steps and Ret. are averages. EpiLoop matches equal-budget forced extra search in accuracy while reducing HCR.

| Agent | Correct | $HCR_c$ | Wrong | $HCR_w$ | Stable+c | Stable+w |
|---|---|---|---|---|---|---|
| standard | 28 | 0.146 | 12 | 0.161 | 14/28 | 5/12 |
| forced-extra | 30 | 0.117 | 10 | 0.133 | 13/30 | 3/10 |
| EpiLoop | 30 | 0.083 | 10 | 0.083 | 15/30 | 5/10 |

Table 7: HCR conditioned on final-answer correctness in the primary intervention comparison. $HCR_c$ and $HCR_w$ are mean HCR for correct and wrong answers. Stable denotes HCR $\leq 0.10$, a descriptive threshold used only for this diagnostic cross-tabulation. The table shows why low HCR cannot be interpreted as correctness: wrong-and-stable trajectories remain possible.

as in the baselines. The safer interpretation is that EpiLoop shifts more trajectories into a stable regime while preserving equal-budget accuracy, but final accuracy and qualitative trajectory taxonomy are still needed to distinguish justified convergence from possible lock-in.

Qualitatively, EpiLoop trajectories differ from standard-agent trajectories in two observable ways. First, when the confidence ceiling or closure audit marks the current state as under-supported, the next query is redirected toward refutation, uncertainty resolution, or independent-source checking rather than another query that restates the current leading hypothesis. Second, source-independence checks make redundant evidence visible as a trajectory problem: the controller can distinguish new documents that add independent support from new documents that repeat the same source group.

Figure 2 shows a concrete instance of this distinction on a FEVER validation item. The claim states that Magic Johnson is the current basketball-operations president of the Los Angeles Lakers; both agents ultimately reach the correct REFUTE answer because Johnson stepped down from that role. The standard agent, however, changes its leading hypothesis several times before the final judgment, producing a high-HCR but eventually correct trajectory. EpiLoop instead keeps the search state stable while issuing refutation-oriented queries, then updates at closure. We include this case not to claim that lower HCR is inherently better, but to show why the trajectory log is useful: two correct final answers can differ substantially in hypothesis dynamics and search control.

### 6.4 Ablations Are Directional, Not Causal Proof

The full EpiLoop condition in this ablation experiment was run with a different random seed than the main RQ3 experiment, producing a reference accuracy of 74.4% rather than 75.0%. Removing the confidence ceiling yields the largest observed accuracy degradation (-6.9pp), followed by removing source-independence checks (-4.4pp) and refutation (-1.9pp). However, confidence intervals overlap substantially across all ablation conditions. The correct interpretation is therefore mechanistic and directional: the confidence ceiling appears to be the most fragile component in this pilot because it prevents high-confidence closure before enough independent evidence has been observed. The HCR values remain close across ablations, suggesting that stability is produced by the overall state-tracking protocol rather than by a single switch.

| Comparison | $\Delta$Acc. | 95% CI | Boot. $p$ | McNemar $p$ | $h$ |
|---|---|---|---|---|---|
| EpiLoop vs forced-extra | +0.000 | [-0.100, +0.100] | 1.0000 | 0.6171 | 0.000 |
| EpiLoop vs standard | +0.050 | [+0.000, +0.125] | 0.2534 | 0.4795 | 0.112 |
| EpiLoop vs cautious | +0.000 | [-0.154, +0.154] | 1.0000 | 0.7237 | 0.000 |

Table 8: Paired accuracy comparisons for the primary intervention experiment. The table is included to make the null result explicit: the current study does not support an accuracy-superiority claim for EpiLoop.

| Trajectory type | Observable audit signature | Diagnostic value for EpiLoop |
|---|---|---|
| Redundant support | Repeated retrieval rounds return the same source groups or the same stance profile. In hover1435, repeated queries retrieve the same three supporting source groups while the leading hypothesis remains support until the final answer step. | This pattern is not necessarily wrong, but it is a source-saturation risk: more retrieval does not add independent evidence. EpiLoop responds by tracking source groups and redirecting toward independent-source checking before closure. |
| Contradiction retrieved but not integrated | Contradictory documents are present in the retrieved set, but the leading hypothesis remains unchanged. In hover4971, contradiction-labeled evidence appears in rounds 2 and 4, yet the final answer remains support with 0.90 confidence. | This separates contradiction retrieval from contradiction integration. EpiLoop responds with refutation intent and blocks closure while contradiction remains unresolved. |
| Premature closure under unresolved uncertainty | The agent reaches a final answer even though the trajectory contains missing-link language, unresolved uncertainty, or insufficient coverage for one claim component. Hover13394 illustrates the target condition: the answer depends on whether a comparison segment is actually grounded. | Closure should depend on whether every claim component is grounded, not only on whether some facts are retrieved. EpiLoop responds with a closure audit over source coverage, contradiction, and missing-link conditions. |

Table 9: Trajectory taxonomy used for qualitative audit. The table does not introduce new aggregate statistics; it translates logged fields (hypothesis, confidence, source groups, stance labels, and closure judgment) into interpretable failure modes that explain why final-answer accuracy alone is insufficient.

## 6.5 Cross-Model Validation

Across all three model families, EpiLoop has the lowest HCR. This supports the claim that the stability effect is not a DeepSeek-only artifact. Cross-model accuracy estimates are strictly exploratory and should not be interpreted as a quantitative comparison across model families. The automatic oracle matcher validates only 15–21 outputs per model-agent condition out of 40 tasks, so the effective sample size for accuracy inference is substantially smaller than $N = 40$.

## 6.6 External Dataset Validation

**Evidence hierarchy.** The external validation changes the evidence hierarchy of the paper. HoVer remains the primary controlled diagnostic experiment because it fixes the evidence pool and directly tests ranking-induced trajectory effects. The FEVER, SciFact, and VitaminC runs instead test transportability: whether the HCR reduction appears when the same EpiLoop controller is applied to other fact-verification task distributions.

**Dataset-specific interpretation.** The strongest external evidence comes from SciFact. Its validation slice has no mixed stances or oracle-document mismatch in the data-quality audit, and EpiLoop reduces HCR from 0.1462 to 0.0304 while matching the standard agent's all-sample accuracy (91.4%). FEVER shows

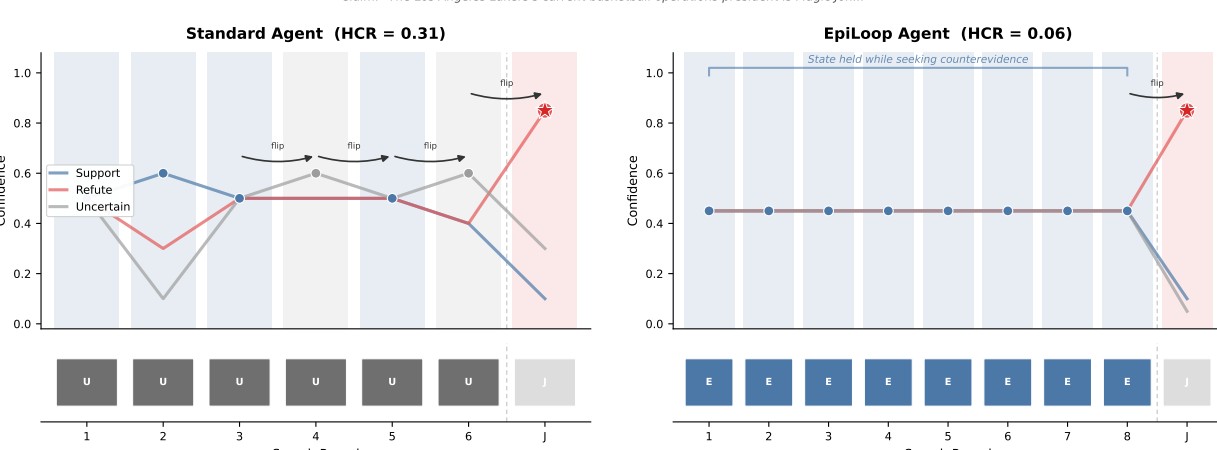

Figure 2: Case-study trajectory visualization on a FEVER claim where both agents reach the correct RE-FUTE answer. The figure illustrates how the diagnostic log separates final correctness from trajectory shape: the standard agent changes its leading hypothesis several times before final judgment, while EpiLoop holds its search state during refutation-oriented evidence gathering and updates at closure. This example is illustrative rather than aggregate evidence; as Table 7 shows, low HCR must be interpreted together with correctness and qualitative trajectory audit.

| Variant | Acc. | 95% CI | ΔAcc. | HCR | Steps | Ret. |
|---|---|---|---|---|---|---|
| full EpiLoop | 74.4 | [59.0, 87.2] | − | 0.081 | 8.0 | 3.0 |
| no closure audit | 74.4 | [59.0, 87.2] | +0.0 | 0.098 | 8.0 | 3.0 |
| no refutation | 72.5 | [57.5, 85.0] | -1.9 | 0.096 | 8.0 | 3.0 |
| no source independence | 70.0 | [55.0, 85.0] | -4.4 | 0.092 | 8.0 | 3.0 |
| no confidence ceiling | 67.5 | [52.5, 82.5] | -6.9 | 0.087 | 8.0 | 3.0 |
| fixed cycle | 75.0 | [60.0, 87.5] | +0.6 | 0.092 | 8.0 | 3.0 |
| random intent | 75.0 | [60.0, 87.5] | +0.6 | 0.092 | 8.0 | 3.0 |

Table 10: EpiLoop ablation on the misleading-top ranking. Removing the confidence ceiling yields the largest observed accuracy degradation, but overlapping confidence intervals mean this should be read as directional pilot evidence.

the same HCR direction but has a high mixed-stance rate, so its accuracy differences should be read as directional. VitaminC is best interpreted as a conflict-heavy stress test: strict oracle accuracy is low for all agents, but EpiLoop still reduces HCR under locally contradictory evidence. We therefore use VitaminC to support the trajectory-stability claim, not to support an accuracy claim.

## 6.7 Cost Trade-off

EpiLoop is not free. Tables 11 and 12 show that the controller lowers HCR at the cost of extra retrieval and token use. In the primary DeepSeek HoVer comparison, EpiLoop and forced extra search both average 3.0 retrieval calls, but EpiLoop uses more tokens. In the non-DeepSeek cross-model runs and the Sprint 7 external-dataset runs, EpiLoop generally uses the full eight-round budget. This is acceptable for a diagnostic actionability probe, but it limits claims about production efficiency.

| Model | Agent | Acc. ($n$) | HCR | Ret. | Tokens |
|---|---|---|---|---|---|
| DeepSeek V4 Pro | standard | 73.7 (19) | 0.151 | 1.3 | 2,588 |
| DeepSeek V4 Pro | EpiLoop | 71.4 (21) | 0.083 | 3.0 | 5,372 |
| DeepSeek V4 Pro | forced-extra | 82.4 (17) | 0.121 | 3.0 | 4,691 |
| Qwen3.6-35B-A3B | standard | 81.0 (21) | 0.125 | 2.3 | 8,675 |
| Qwen3.6-35B-A3B | EpiLoop | 84.2 (19) | 0.028 | 8.0 | 54,947 |
| Qwen3.6-35B-A3B | forced-extra | 95.0 (20) | 0.058 | 8.0 | 32,793 |
| GLM-5.1 | standard | 88.2 (17) | 0.209 | 2.4 | 6,532 |
| GLM-5.1 | EpiLoop | 88.2 (17) | 0.034 | 8.0 | 30,999 |
| GLM-5.1 | forced-extra | 86.7 (15) | 0.086 | 8.0 | 22,163 |

Table 11: Cross-model validation and cost profile on the same 40-task slice. Parentheses in the Acc. column give the number of outputs matched by the automatic oracle matcher. Accuracy is exploratory; HCR is computed from logged trajectories. EpiLoop has the lowest HCR in each model family, but it also uses more retrieval and tokens than the standard agent.

| Dataset | Agent | Acc.[†] | Acc.all | HCR | Ret. | Tokens |
|---|---|---|---|---|---|---|
| FEVER | standard | 77.8 | 41.2 | 0.1140 | 6.6 | 13,079 |
| FEVER | EpiLoop | 90.0 | 52.9 | 0.0588 | 8.0 | 17,380 |
| FEVER | forced-extra | 76.9 | 58.8 | 0.0956 | 8.0 | 14,964 |
| SciFact | standard | 97.0 | 91.4 | 0.1462 | 3.2 | 7,508 |
| SciFact | EpiLoop | 100.0 | 91.4 | 0.0304 | 8.0 | 23,560 |
| SciFact | forced-extra | 100.0 | 82.9 | 0.0429 | 8.0 | 20,103 |
| VitaminC | standard | 58.3 | 33.3 | 0.1318 | 4.6 | 12,680 |
| VitaminC | EpiLoop | 53.8 | 33.3 | 0.0506 | 8.0 | 25,394 |
| VitaminC | forced-extra | 53.8 | 33.3 | 0.1131 | 8.0 | 22,470 |

Table 12: External dataset validation with DeepSeek V4 Pro. Acc.[†] excludes predictions classified as uncertain; Acc.all counts uncertain as incorrect. VitaminC accuracy is a noisy secondary outcome because the validation slice contains mixed stances and claim-variant mismatch. HCR is computed directly from trajectory logs.

## 7 Discussion

### 7.1 What the Null Accuracy Result Means

The experiments show that epistemic lock-in is measurable as a trajectory-level reliability phenomenon. Retrieval improves accuracy, but document order can still change final hypotheses. Equal-budget extra search improves accuracy to the same level as EpiLoop, but does not reduce HCR as much. Generic caution increases answer divergence and HCR rather than cleanly stabilizing the trajectory. These patterns support the diagnostic premise: final-answer accuracy alone hides important differences in how iterative agents search, revise, and stop.

The null accuracy result is informative. In a controlled evidence setting, once relevant evidence is available and the retrieval budget is matched, explicit epistemic-state tracking may not add new facts. Instead, it changes the trajectory: when the agent searches, what kind of evidence it asks for, and how stable its hypothesis remains. This explains why EpiLoop can reduce HCR without outperforming forced extra search on final accuracy. However, Table 7 also shows that low HCR is not automatically desirable. It can correspond to correct convergence or wrong stability. HCR is therefore not interchangeable with accuracy; it captures a distinct dimension of trajectory quality that must be interpreted alongside contradiction exposure, source independence, closure behavior, and final answers.

| Dataset | Comparison | ΔHCR | 95% CI | $p$ | $d$ |
|---------|-----------|------|--------|-----|-----|
| FEVER | standard − EpiLoop | +0.0552 | [+0.0120, +0.1026] | 0.0084 | +0.584 |
| FEVER | EpiLoop − forced-extra | -0.0368 | [-0.0735, -0.0037] | 0.0456 | -0.494 |
| SciFact | standard − EpiLoop | +0.1158 | [+0.0751, +0.1568] | <.001 | +0.940 |
| SciFact | EpiLoop − forced-extra | -0.0125 | [-0.0250, -0.0036] | 0.0096 | -0.347 |
| VitaminC | standard − EpiLoop | +0.0812 | [+0.0270, +0.1412] | <.001 | +0.609 |
| VitaminC | EpiLoop − forced-extra | -0.0625 | [-0.0982, -0.0268] | <.001 | -0.743 |

Table 13: Paired HCR comparisons in the external validation runs. Positive values in standard − EpiLoop mean the standard agent has higher HCR; negative values in EpiLoop − forced-extra mean EpiLoop has lower HCR than forced extra search. These results support HCR transportability, not a universal accuracy-improvement claim.

## 7.2 What Cross-Dataset Transportability Means

The external validation strengthens a different claim from accuracy improvement. Across FEVER, SciFact, and VitaminC, EpiLoop consistently reduces HCR relative to both the standard agent and forced extra search. This pattern suggests that explicit epistemic-state tracking affects a portable trajectory property: the rate at which the leading hypothesis changes over the search loop. The effect appears in a clean scientific fact-verification setting (SciFact), a smaller mixed-stance FEVER slice, and a deliberately conflict-heavy VitaminC setting.

The accuracy evidence is weaker. On SciFact, EpiLoop matches standard all-sample accuracy while lowering HCR. On FEVER, EpiLoop has higher accuracy when uncertain outputs are excluded, but many validation items contain mixed stances. On VitaminC, strict oracle accuracy is not a reliable indicator of model reasoning quality because locally contradictory evidence can make oracle-supported examples appear unsupported. We therefore treat the cross-dataset results as evidence for HCR transportability, not as evidence that EpiLoop is a general-purpose accuracy improver.

## 7.3 Practical Use of Trajectory Diagnostics

The diagnostic protocol is designed for offline analysis, but it also suggests an operational use. In a deployed search agent, HCR can be monitored as a trajectory-health signal. When HCR is unusually low while confidence is high but source coverage is narrow, the system may be entering lock-in and could trigger a human review, a mandatory refutation query, or a source-diversification step. Conversely, when HCR is high but accompanied by successful contradiction integration and evidentially justified closure, the system may be exhibiting healthy revision rather than instability.

This monitoring pattern does not require per-task ground truth. It requires only the behavioral proxies already tracked by EpiLoop. That makes it compatible with settings where labels are unavailable but trajectory logs exist, such as legal research, scientific literature search, or enterprise document QA. We do not evaluate this deployment pattern in the current study. The contribution is the measurement scaffolding needed to ask whether such monitoring improves reliability.

## 7.4 Why State-Aware Refutation Is Different from More Search

One might ask whether the result is simply an expensive version of "search more." The equal-budget forced-extra-search baseline is designed to test that possibility. The accuracy tie suggests that extra search is a strong baseline. The HCR difference suggests that state-aware search changes the shape of the trajectory, not merely its length. Forced extra search spends budget uniformly. EpiLoop spends budget conditionally, using confidence, source coverage, and contradiction status to decide whether to seek independent evidence, refute the current leader, or close.

This distinction matters for reliability audits. A system that searches more can still be locked into a narrow evidence neighborhood. A system that audits state can ask whether new evidence is independent, whether

contradiction has been integrated, and whether confidence is justified by coverage. The present study does not prove that this state-aware policy is optimal. It shows that the difference is measurable.

## 8 Related Work

### 8.1 Retrieval-Augmented and Iterative Search Agents

Retrieval-augmented generation grounds language-model outputs in retrieved evidence (Lewis et al., 2020). Iterative variants such as IRCoT, ReAct, Self-RAG, and active retrieval systems interleave reasoning with retrieval or critique (Trivedi et al., 2022; Yao et al., 2022; Asai et al., 2023; Jiang et al., 2023). These systems show that multi-step retrieval can improve knowledge-intensive reasoning, but evaluation often focuses on final accuracy, retrieval success, or citation quality. EpiLoop instead evaluates how hypotheses evolve across the loop.

### 8.2 Corrective and Adaptive RAG

Recent RAG systems increasingly make retrieval conditional. Self-knowledge guided retrieval asks a model to decide when external knowledge is needed (Wang et al., 2023); Adaptive-RAG routes questions to retrieval strategies based on predicted question complexity (Jeong et al., 2024); CRAG evaluates retrieval quality and triggers corrective actions such as web search or document decomposition (Yan et al., 2024). RAG evaluation work similarly emphasizes retrieval quality, faithfulness, answer correctness, and robustness of the generated response (Chen et al., 2023; Es et al., 2023; Gao et al., 2023).

EpiLoop is complementary to these systems. It does not propose a new retriever, router, or corrective generation pipeline. Its diagnostic target is the *trajectory*: whether the agent's current hypothesis shapes later queries, whether contradiction is integrated, whether source coverage is independent, and whether closure is justified. A corrective RAG method can be evaluated with EpiLoop-style logs, and EpiLoop's controller can be viewed as a minimal actionability probe rather than a full adaptive RAG architecture.

### 8.3 Self-Correction, Reflection, and Process Supervision

Self-consistency, Self-Refine, Reflexion, and Tree of Thoughts improve reasoning by sampling, critiquing, revising, or branching over reasoning paths (Wang et al., 2022; Madaan et al., 2023; Shinn et al., 2023; Yao et al., 2023). EpiLoop is complementary. It does not propose a new deliberation algorithm; it instruments the epistemic state of an existing iterative search loop and asks whether state-aware interventions reduce lock-in symptoms.

Several studies have shown that LLM self-correction can fail when the model lacks external grounding, leading to repeated errors or confidence inflation rather than genuine improvement (Huang et al., 2024; Stechly et al., 2023). These findings motivate why EpiLoop does not rely on the agent to self-correct through introspection alone. Instead, it uses external retrieval under a state-aware policy to supply independent evidence that the agent may not spontaneously seek.

Process supervision demonstrates that step-level verification signals can improve reasoning reliability beyond final-answer feedback alone (Lightman et al., 2023). EpiLoop shares the intuition that intermediate states matter, but applies it to multi-round retrieval rather than chain-of-thought math reasoning.

### 8.4 Uncertainty, Ranking Effects, and Relevance Feedback

Prior work on language-model uncertainty asks whether models know what they know and how confidence can be estimated or calibrated (Kadavath et al., 2022; Lin et al., 2023; Tian et al., 2023). EpiLoop uses confidence differently: as a control signal for stopping and refutation, interpreted alongside source coverage and source independence. Work on long-context use and "lost in the middle" effects shows that position and ranking influence generation quality (Liu et al., 2023). EpiLoop extends this concern to multi-round search agents, where ranking affects not only one generation step but the trajectory of future queries and hypotheses.

Classical relevance feedback offers another connection (Robertson & Sparck Jones, 1976; Rocchio, 1971). In relevance feedback, retrieved documents refine subsequent queries. EpiLoop applies a similar control idea to epistemic state rather than topical relevance: the controller refines queries not because documents are off-topic, but because the current evidence profile indicates lock-in risk.

### 8.5 Agent Reliability

Survey work on LLM-based autonomous agents highlights open challenges in planning, memory, tool use, and self-evolution (Wang et al., 2024). OS-Copilot shows that agents can self-improve through interaction traces (Xie et al., 2024). EpiLoop complements these directions by making epistemic trajectory quality observable. When agents self-correct, it matters whether they revise toward truth or toward self-consistency with an early commitment. The tracking approach introduced here provides a starting point for measuring that distinction.

## 9 Limitations

The primary controlled study uses 40 HoVer-derived tasks, so confidence intervals for accuracy differences remain wide. HoVer is well suited to support/refute evidence and source-group reasoning, but the conclusions may not transfer directly to open-domain web search, long-form synthesis, ambiguous QA, or summarization. Retrieval is simulated over a frozen local corpus, which supports causal diagnosis but does not capture live search failures, search-engine drift, snippet effects, SEO, or recency bias.

The external validation also has limits. The additional datasets are validation slices rather than benchmark-scale evaluations. SciFact is the cleanest external dataset, but it is still a fact-verification benchmark rather than a broad search environment. VitaminC contains systematic claim-variant mismatch, mixed stances, and severe label imbalance; we therefore treat its strict accuracy as noisy and use it mainly as a stress test for trajectory stability under conflicting evidence. Final-answer accuracy in the external runs depends on a deterministic parser that maps free-form answers to SUPPORTED, REFUTED, or UNCERTAIN; parser errors or conservative uncertainty detection can change reported accuracy.

HCR is a proxy. Low HCR can mean justified convergence or unjustified lock-in, and high HCR can mean healthy revision or unstable reasoning. Cross-model accuracy is exploratory because the current automatic oracle matcher validates only 15–21 outputs per condition. EpiLoop increases inference cost and uses fixed thresholds rather than learned policies.

An additional limitation concerns the refutation-intent mechanism. EpiLoop's rule-based policy triggers refutation queries only when the leading hypothesis has been stable for $\kappa$ steps while contradiction evidence is present. A simpler alternative, periodically issuing a refutation query every $n$ steps regardless of epistemic state, was not evaluated. Such a schedule-driven baseline would be easier to implement and might achieve comparable HCR reduction if refutation timing matters less than refutation frequency. Distinguishing state-aware from schedule-driven refutation requires a dedicated ablation.

## 10 Broader Impact

Trajectory diagnostics for search agents can support safer deployment in settings where users depend on evidence synthesis: legal research, medical literature review, compliance search, scientific discovery, and education. The positive use case is not automatic truth certification. It is better auditability. A system that logs hypothesis changes, source coverage, contradiction integration, and closure decisions gives human reviewers a clearer view of how an answer was reached.

There are also risks. A low HCR score could be misinterpreted as correctness, even though it may indicate either justified convergence or lock-in. Conversely, an adversary could attempt to manipulate retrieved evidence so that the agent appears stable while suppressing contradiction. The same mechanisms used to detect lock-in could be inverted to induce it, for example by steering retrieval toward redundant support.

These risks reinforce the need to report trajectory diagnostics as evidence for human review, not as automatic guarantees.

## 11    Conclusion

We introduced epistemic lock-in as a measurable reliability failure in iterative search agents. The main argument is that search-agent evaluation should look beyond retrieved context and final accuracy to the agent's evolving epistemic trajectory: what hypotheses it maintains, whether it integrates contradiction, whether source evidence is independent, and whether closure is justified.

EpiLoop operationalizes this view as a lightweight inference-time actionability probe. In a controlled HoVer-derived study, EpiLoop does not outperform equal-budget forced extra search in final accuracy. Instead, it reduces hypothesis instability: it achieves the lowest HCR in the primary study and shows consistent HCR reduction across DeepSeek V4 Pro, Qwen3.6-35B-A3B, GLM-5.1, and additional fact-verification datasets. The result is a modest but useful reliability claim. Epistemic-state tracking may not make an agent more accurate than simply searching more, but it can make the search process more stable, inspectable, and diagnostically meaningful across several evaluation settings.

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

## A   Reproducibility Details

The main experiments use a frozen local evidence pool derived from HoVer. Each task contains a claim, the oracle label, a list of documents, source-group metadata, stance annotations, and ranking variants. The corrected source groups are Wikipedia page titles rather than synthetic document identifiers. This correction is important because source independence is only meaningful when repeated evidence can be traced to repeated source groups.

All RQ1–RQ4 conditions are run on the same 40 tasks. RQ1 evaluates 13 conditions: one closed-book baseline and 3 loop modes across 4 rankings. RQ2 evaluates ranking perturbations across contradiction timing. RQ3 evaluates five agents under an equal max-round budget. RQ4 evaluates seven EpiLoop ablation variants. The full corrected run contains 1,476 valid records after four API errors in the original 1,480-call run.

## B   Additional Statistical Details

The paired EpiLoop versus forced-extra comparison has two discordant pairs in each direction: EpiLoop wrong / forced-extra correct = 2, and EpiLoop correct / forced-extra wrong = 2. The paired bootstrap accuracy difference is exactly 0.000 with 95% CI [-0.100, +0.100]. The McNemar statistic is $\chi^2(1) = 0.250$, $p = 0.6171$.

The paired EpiLoop versus standard comparison has an observed +5.0pp accuracy difference with 95% CI [0.000, +0.125], bootstrap $p = 0.2534$, and McNemar $p = 0.4795$. This comparison is directionally positive but not statistically significant at the current sample size.

For HCR, Cohen's $d$ is 0.5471 for EpiLoop versus standard and 0.3805 for EpiLoop versus forced extra search. We report these as effect-size estimates rather than significance claims.

## C   Cross-Model Accuracy Caveat

Cross-model HCR is computed from trajectories and does not require final-answer oracle matching. Cross-model accuracy, however, depends on an automatic oracle matcher that recognizes whether free-form model outputs correspond to the HoVer support/refute label. The matcher validates only 15–21 outputs per model-agent condition out of 40 tasks. Accuracy values in the cross-model run are therefore useful as sanity checks but should not be used as primary evidence.

## D   External Dataset Accuracy Caveat

The external-dataset runs use strict oracle scoring for comparability: SUPPORTED, REFUTED, and UN-CERTAIN predictions are mapped from the final free-form answer, and uncertain predictions are either excluded from Acc.$^\dagger$ or counted as incorrect in Acc.all. Under this rule, a VitaminC item with oracle label SUPPORTED is counted as incorrect if the model answers that the evidence is contradictory, insufficient, or not supported. We keep this rule because relabeling individual outputs after inspection would make cross-agent comparisons unstable.

The interpretation is narrower than the scoring rule. VitaminC contains systematic local-evidence ambiguity in this validation slice: items include mixed support/refute stances and a claim-variant mismatch, while the oracle labels are strongly skewed toward SUPPORTED. For that reason, low VitaminC strict accuracy should not be treated as clean evidence of reasoning failure, nor should small differences in VitaminC strict accuracy be treated as agent superiority. We use VitaminC primarily to test whether trajectory diagnostics behave sensibly under adversarially conflicting evidence.

## E   Example Trajectory Reading

The trajectory audit reads each step as a tuple of hypothesis, confidence, retrieved document IDs, source groups, stance labels, and closure judgment. A typical lock-in-consistent pattern has the following form: the

initial hypothesis is support; the next two queries retrieve evidence from the same source group; confidence increases; a contradictory document appears but is summarized as background; the agent stops because the answer appears sufficiently supported. EpiLoop flags this trajectory because confidence is high, source coverage is narrow, and contradiction remains unresolved.

The important point is not the final label alone. If the final answer is correct but the trajectory reached it through redundant evidence and unexamined contradiction, the trajectory remains diagnostically risky. Conversely, an incorrect answer with high HCR may indicate productive uncertainty rather than lock-in. The protocol is intended to support this finer-grained audit.

## F   Ethics and Data Availability

The study uses the public HoVer benchmark and does not involve human subjects, private data, or personally identifiable information. All experiments are inference-only. Code, task schemas, corrected task slices, result JSONL files, and analysis scripts are intended for release with the submission package after anonymization.

