# OpenReview forum: "EpiLoop: A Diagnostic Protocol for Measuring Epistemic Lock-in in Iterative Search Agents"
_TMLR — Under review for TMLR_

### Review · Reviewer_bHgp · 2026-06-25

**Summary Of Contributions:**

Note: I am not from the agent or RAG community, and my confidence in judging the paper’s positioning relative to the most recent agent-evaluation literature is therefore relatively low. My assessment is mainly from a general machine learning / evaluation-methodology perspective.

This paper studies a failure mode that the authors call epistemic lock-in in iterative search agents. The central idea is that an early hypothesis may influence later query generation, evidence selection, confidence estimation, and stopping decisions, causing additional retrieval to reinforce rather than revise the current answer. To study this, the paper proposes a diagnostic protocol based on trajectory-level metrics. The primary metric is Hypothesis Change Rate (HCR), defined as the fraction of adjacent trajectory steps in which the leading hypothesis changes. The paper also discusses ranking sensitivity, contradiction integration, source independence, closure behavior, and cost. The paper further proposes EpiLoop, a lightweight inference-time wrapper/controller around an iterative search agent. The main controlled experiment is a 40-task HoVer-derived setup with a frozen evidence pool and manipulated rankings. The headline result is that EpiLoop matches equal-budget forced extra search in final accuracy, 75.0% vs. 75.0%, while reducing HCR from 0.121 to 0.083.

I appreciate the paper’s motivation that final-answer accuracy alone may miss important trajectory-level behavior in iterative search systems. I also appreciate that the paper is relatively honest about the null accuracy result and explicitly states that EpiLoop does not outperform equal-budget extra search in final-answer accuracy. However, I find the overall contribution limited for TMLR in its current form. The proposed controller is rule-based and relatively simple, the main experiment is small, the central metric has substantial construct-validity ambiguity, and the empirical evidence mainly shows a reduction in HCR rather than a clear improvement in reliability or correctness.

**Audience:**

Yes

**Audience Explanation:**

Some researchers working specifically on RAG agents, iterative search systems, and agent-evaluation diagnostics may find the topic interesting. The paper raises a reasonable concern: final-answer accuracy can hide differences in search trajectories, and iterative agents may reinforce early hypotheses rather than revise them. This is a relevant issue for agent reliability.

**Broader Impact Concerns:**

N/A in this case.

**Claims And Evidence:**

No

**Claims Explanation:**

The narrowest claims are supported: in the reported settings, EpiLoop reduces HCR relative to standard iteration and equal-budget forced extra search, while not improving final accuracy over forced extra search. But the paper is transparent about this result. The main HoVer-derived experiment reports exactly tied accuracy between EpiLoop and forced extra search, and the external validation slices are also presented cautiously as HCR-transportability checks rather than benchmark-scale accuracy claims.

However, I do not find the evidence convincing for the broader framing around epistemic lock-in. The main issue is construct validity. The paper defines epistemic lock-in as a path-dependent reliability failure, but the main empirical signal is reduced HCR. Reduced HCR does not by itself imply less lock-in. In fact, a locked-in trajectory could have very low HCR. The paper does include some discussion of this ambiguity and states that HCR should be interpreted alongside correctness, contradiction integration, source coverage, and closure behavior. Nevertheless, the quantitative evidence still centers on HCR reduction, and the additional diagnostics are not developed strongly enough to establish that EpiLoop reduces the harmful form of lock-in rather than merely reducing hypothesis changes.

**Requested Changes:**

- The current paper relies heavily on HCR reduction, but HCR is ambiguous. Low HCR can be good convergence or bad lock-in; high HCR can be instability or healthy revision. The paper should not present HCR reduction as strong evidence of reduced epistemic lock-in unless it is paired with stronger evidence about contradiction integration, source independence, closure quality, and wrong-but-stable trajectories.
- The paper should quantify cases such as low-HCR wrong answers, high-confidence closure with narrow source coverage, retrieved contradictions that are not integrated, and cases where the agent remains stable despite counter-evidence. These are much closer to the claimed failure mode than HCR alone.
- A simple baseline that issues a refutation query every n steps, or before final closure, is necessary. Without this baseline, it is difficult to know whether EpiLoop’s effect comes from epistemic-state tracking or simply from injecting periodic refutation / independent-source queries.
- The 40-task HoVer-derived study is too small to support strong claims. A larger controlled study and more systematic external evaluation would be needed. The current external datasets are presented as slices, and several accuracy results are parser-dependent or explicitly noisy.
- The current figures are **NOT** publication-quality in my view. Figure 1 is too dense and difficult to read, while Figure 2 looks rough and insufficiently polished. Some visual elements appear AI-generated or mechanically assembled without careful refinement. The authors should redraw the figures in a cleaner, more standard scientific style and ensure that each figure directly supports an empirical or methodological point.
- The paper should better explain why this is a TMLR-level machine learning contribution rather than a specialized RAG/agent diagnostic protocol. At present, the contribution seems narrow, exploratory, and primarily rule-based.

---

### Review · Reviewer_aDJt · 2026-07-06

**Summary Of Contributions:**

This paper proposed a new possible issue in the agents, which is called epistemic lock-in. It's a feedback pathology. They assume the agent first has an assumption, and all the following steps will be based on the assumption. Then if the first assumption is wrong, the later steps will all be wrong. This concept is first introduced in the psychology paper.

To analyze whether the agent has the epistemic lock-in issue, this paper proposes a new lightweight diagnostic framework EpiLoop. The framework checks the trajectory, including leading hypotheses, hypothesis changes, source groups, contradiction exposure, confidence changes, and stopping behavior. Hypothesis Change Rate (HCR) is proposed to determine whether the leading hypothesis is stable or changes frequently. EpiLoop is added during the inference time.

The paper also conducts experiments on several datasets. However, this lightweight EpiLoop can't improve the final performance (accuracy).

**Audience:**

No

**Audience Explanation:**

This paper introduces a new way to diagnose the agent. However, by far, I don't think the findings are ready for most of the audience.

**Broader Impact Concerns:**

No concerns regarding the ethical implications of the work.

**Claims And Evidence:**

No

**Claims Explanation:**

The idea of epistemic lock-in is interesting, however, my main concern is that the experiment is not clear enough to prove this concept.

The paper's central claims are:

1) lock-in is measurable via HCR and related proxies;

2) EpiLoop reduces HCR at matched budget, demonstrating that lock-in is actionable;

3) these diagnostics are useful for reliability auditing.

I believe the current evidence does not support any of the three, for reasons that are largely about measurement validity rather than effect size.

First, the hypothesis-extraction protocol, which is the paper's core measurement instrument, is unspecified and likely unequal across conditions. Per-step hypotheses (hence HCR and flip rates) must be extracted from model outputs, but the paper never specifies how. EpiLoop conditions use structured state fields; baselines apparently rely on parsing free-form text (Sec. 4, Sec. 5.4, App. C). Either baselines were also forced to emit structured hypotheses (an intervention in itself) or their HCR is parser-derived while EpiLoop's is not — a measurement-mechanism confound that by itself could produce the headline HCR gap. No prompt templates, parsing rules, or human-agreement validation are provided. Figure 2's legend ("logged or inferred controlled intent") suggests some states are post-hoc inferred.

Another issue is the three-label hypothesis space cannot represent the epistemic states the paper claims to study. HoVer/FEVER/VitaminC items with mixed stances or claim-variant mismatch (which the paper's own audit shows are systematic in VitaminC and present in FEVER) admit conditional judgments ("supported under reading A, refuted under reading B") as the epistemically correct output. The deterministic parser collapses such outputs into support/refute/uncertain, conflating them with ignorance. This pollutes accuracy (acknowledged) but also HCR (not acknowledged): a stable conditional belief whose phrasing emphasizes different branches across rounds registers as repeated flips in baselines, while EpiLoop's forced single-label output format suppresses conditional expression at the source. The oracle matcher validating only 15 to 21 over 40 outputs in cross-model runs is direct evidence that a large fraction of real outputs do not fit the label space. A paper arguing that accuracy is too coarse a measure of epistemic state itself represents epistemic state with three symbols.

Moreover, the HCR reduction is plausibly a mechanical artifact of the intervention, and the paper's own ablations indicate state-awareness is not the active ingredient. The controller's design holds the hypothesis fixed during refutation-oriented rounds and updates at closure (Fig. 2), i.e., it suppresses intermediate flips by construction; measuring reduced flips then partially measures the intervention's definition. However, in Table 10: fixed cycle and random intent, the experiment ignores epistemic state entirely, and achieves HCR 0.092 vs. full EpiLoop's 0.081 and slightly higher accuracy, while forced-extra sits at 0.121. Nearly the entire stability effect is reproduced by structured intent rotation without any state tracking. This directly undercuts Sec. 7.4's central argument and the "actionability of epistemic state". The paper does not discuss this. Additionally, HCR is normalized by trajectory length; EpiLoop nearly always exhausts the 8-round budget while baselines terminate earlier (Tables 11–12), mechanically diluting flips. No length-controlled analysis is given.

Another issue is that the ranking-perturbation protocol (RQ2) is underspecified and confounded with multi-hop dependency structure. The construction of the four rankings and three contradiction-timing conditions is never defined. More fundamentally, Algorithm 1 specifies query-conditioned retrieval, yet controllable "contradiction timing" implies an exposure-stream design; the paper does not resolve which was implemented, and the two readings have opposite implications. Under an exposure stream, HoVer's many-hop dependency chains mean a downstream document surfaced before its bridging context cannot be recognized as relevant by any agent. This can kind of explain the observed 10–20% flip rates because it is theoretically impossible due to the setup, not agent pathology (and it is unstated whether consumed documents can be revisited, which would asymmetrically favor EpiLoop's longer trajectories). Under query-driven retrieval, the manipulated orderings cannot occur in deployment because dependency order is endogenously enforced by the query process, so RQ2 measures a counterfactual with no external referent. Either way, the paper's primary evidence for lock-in is compromised. At N=40, the "non-monotonic" timing pattern (differences of 2 to 4 tasks) is also fully consistent with noise.

Last but not least, the diagnostic utility is asserted, never evaluated. Sec. 7.3 proposes monitoring low-HCR/high-confidence/narrow-coverage as a lock-in flag, but no discrimination analysis is performed. Table 7 shows that HCR is identical for both with and without EpiLoop trajectories (0.083 vs. 0.083). If the HCR is a diagnostic metric, how can we interpret it or use it? It seems that given different HCR, the agent can still be correct or wrong.

**Requested Changes:**

1. Fully specify and validate the hypothesis-extraction protocol: identical extraction across all conditions, prompt templates and parsing rules, and human-agreement rates with a confusion matrix that includes conditional/compound answers as a category.

2. Report the fraction of raw outputs per condition that fail to map into the three-label space, and analyze what fraction of baseline HCR flips co-occur with conditional/hedged phrasing or low parser confidence.

3. Directly address the fixed-cycle/random-intent results: either demonstrate a measurable advantage of state-conditioning over schedule-driven intent rotation, or revise the paper's claims from "epistemic-state tracking reduces instability" to "structured intent rotation reduces measured flips".

4. Provide a length-controlled HCR analysis (or absolute flip counts at matched trajectory length).

5. Specify the retrieval implementation precisely (exposure stream vs. query-conditioned; ranking-injection mechanism; per-round k; document revisitability), give construction pseudocode for all rankings/timings, and report what fraction of RQ2 flips occur on tasks whose multi-hop dependency chain was inverted by the perturbation, or add an oracle-dependency-order control.

6. More discussion on how to use/interpret the HCR score.

---

### Review · Reviewer_qviH · 2026-07-15

**Summary Of Contributions:**

This paper studies epistemic lock-in in iterative search agents, a path-dependent failure mode in which an early hypothesis influences subsequent queries, evidence selection, confidence, and stopping decisions. It introduces a controlled diagnostic protocol that fixes the evidence pool while perturbing document ranking and contradiction timing, with Hypothesis Change Rate (HCR) as its primary trajectory-level metric. The paper also proposes EpiLoop, an inference-time controller that tracks hypotheses, confidence, source coverage, closure status, and refutation intent to guide subsequent searches. On a 40-task HoVer-derived evaluation, EpiLoop matches equal-budget forced extra search in accuracy (75.0%) while achieving lower HCR (0.083 versus 0.121). Cross-model experiments and validation slices from FEVER, SciFact, and VitaminC show similar reductions in HCR, although accuracy effects are inconsistent and parser-dependent.

Strengths：
1. The paper identifies an important limitation of evaluating iterative search agents solely through final-answer accuracy and makes a compelling case for examining the evolution of hypotheses, evidence use, and stopping decisions throughout the search trajectory.
2. It provides a coherent diagnostic framework for epistemic lock-in and carefully distinguishes this phenomenon from related but narrower failures, including retrieval absence, position bias, redundant evidence, contradiction non-integration, and premature closure.
3. The paper connects diagnosis with intervention by introducing EpiLoop, a lightweight controller that makes the agent’s epistemic state explicit and uses it to guide refutation, source diversification, and closure decisions without modifying the underlying model or retriever.
4. The empirical design includes meaningful controls—particularly fixed-evidence ranking perturbations and an equal-budget forced-search baseline—and supplements the primary experiment with cross-model and cross-dataset evaluations. This provides a reasonably broad assessment of the proposed trajectory-level effect.

Weaknesses:
1. The central proxy has a substantial construct-validity limitation. A low HCR may reflect justified convergence, delayed commitment, or persistent commitment to an incorrect hypothesis. Although the paper acknowledges this ambiguity, it limits the extent to which lower HCR can validate the broader framing of HCR as a diagnostic proxy for epistemic lock-in.
2. The empirical evaluation does not sufficiently validate the mechanisms underlying EpiLoop. More direct measures—such as contradiction integration, source independence, and justified closure—are defined but not systematically reported, while the fixed-cycle and random-intent ablations perform similarly to the full controller.
3. The practical benefit of EpiLoop is unclear given its computational overhead. It often uses the full search budget and substantially more tokens than the standard agent; it also consumes more tokens than the equal-retrieval forced-extra baseline without improving final-answer accuracy. The paper does not provide a cost-normalized evaluation showing whether the reduction in HCR justifies this additional expense.
4. The auxiliary validation remains limited. Cross-model accuracy relies on an automatic oracle matcher that recognizes only 15–21 of the 40 outputs per condition, while the external-dataset experiments use relatively small validation slices and a deterministic answer parser that may introduce classification errors. These limitations weaken the accuracy conclusions and the generalizability of the auxiliary results.

**Audience:**

Yes

**Audience Explanation:**

The paper addresses a relevant problem for researchers working on retrieval-augmented generation, iterative search agents, and AI reliability: final-answer accuracy may hide important differences in how agents gather evidence, revise hypotheses, and decide when to stop.

Even though the connection between HCR and epistemic lock-in requires stronger validation, the paper’s trajectory-level perspective, diagnostic setup, and finding that lower HCR does not necessarily improve accuracy provide useful directions for future evaluation of search agents.

**Broader Impact Concerns:**

I do not identify major broader-impact concerns beyond those already discussed in the paper.

**Claims And Evidence:**

No

**Claims Explanation:**

The experiments support the narrower claim that EpiLoop reduces HCR, but they do not convincingly show that it reduces epistemic lock-in. Low HCR can indicate either justified convergence or an incorrect hypothesis that remains unchanged. Table 7 indeed shows that several incorrect EpiLoop trajectories are classified as stable. Therefore, HCR alone cannot distinguish improved epistemic behavior from erroneous lock-in or delayed commitment.

The paper defines more direct measures, such as contradiction integration, source independence, and closure quality, but does not report them systematically in the main experiments. Some HCR comparisons are also affected by differences in trajectory length, while the primary evaluation contains only 40 tasks.

The experimental results are clearly presented, and the authors appropriately acknowledge the null accuracy result and other limitations. Nevertheless, the current evidence supports reduced hypothesis-change frequency rather than the broader claim of measuring or mitigating epistemic lock-in.

**Requested Changes:**

1. Provide stronger evidence that HCR actually captures epistemic lock-in. A low HCR may occur because the agent correctly converges, remains stuck on an incorrect answer, or simply postpones changing its stated hypothesis until the final step. The evaluation should distinguish these cases before interpreting lower HCR as evidence of better epistemic behavior.

2. Systematically report the more direct indicators introduced in the paper. Contradiction integration, source independence, evidence redundancy, and whether the agent stops with sufficient evidence should be measured across all main experimental conditions, rather than illustrated mainly through selected examples. These results should also be separated according to whether the final answer is correct and whether contradictory evidence was retrieved.

3. Show that the improvement comes specifically from using the tracked state. The fixed-cycle and random-intent variants achieve results similar to the full EpiLoop controller. The authors should add a baseline that issues refutation queries on a fixed schedule and compare all variants using the same numbers of search rounds, retrieval calls, and tokens. This would clarify whether EpiLoop benefits from state-dependent decisions or simply from performing more diverse searches.

4. Make the HCR comparisons fair across trajectories of different lengths. Because HCR divides the number of hypothesis changes by the number of adjacent steps, longer trajectories may obtain lower values even without better behavior. The paper should also report the total number of hypothesis changes, results computed over the same number of steps, when the first hypothesis is formed, and how quickly the agent revises after encountering counterevidence.

---

### Review · Reviewer_L3Fa · 2026-07-18

**Summary Of Contributions:**

The paper defines epistemic lock-in as a trajectory-level reliability failure in iterative search agents and introduces a diagnostic protocol centered on Hypothesis Change Rate (HCR) to quantify hypothesis instability along an agent’s search trajectory. The authors propose EpiLoop, a lightweight inference-time controller that tracks five epistemic-state variables (hypothesis, confidence, source coverage, refutation intent, and closure audit) and uses simple rules to guide search intent. In controlled HoVer-derived experiments and external validation slices (FEVER, SciFact, VitaminC), EpiLoop consistently reduces HCR relative to standard iteration and an equal-budget forced-extra-search baseline, while final-answer accuracy remains statistically indistinguishable from forced extra search in the main study—supporting the claim that trajectory diagnostics expose stability properties not captured by accuracy alone.

**Audience:**

Yes

**Audience Explanation:**

1. The paper articulates and formalizes “epistemic lock-in” as a loop-level pathology distinct from single-shot position bias or plain retrieval errors, bridging concepts from cognitive science (confirmation bias, belief perseverance) to iterative retrieval agents.

2. HCR is a simple, auditable, and model-agnostic proxy for hypothesis stability; pairing it with ranking perturbations and source-coverage checks yields an interpretable diagnostic toolkit.

3, EpiLoop is an actionable probe that minimally intervenes (rule-based intents, closure audit, confidence ceilings) to test whether state tracking can affect trajectories without introducing new models or training.

**Claims And Evidence:**

No

**Claims Explanation:**

1. It's quite obvious that the main image was generated by AI, containing some perplexing nodes and meaningless arrow colors.

2. low HCR can indicate justified convergence or unjustified lock-in; although the paper acknowledges this, stronger operational links between HCR and independently adjudicated “epistemic quality” would improve construct validity.

3. Thresholds like confidence ceiling, closure audit, refutation triggers, appear fixed but their selection rationale and sensitivity are limited; results could be brittle to such choices.

**Requested Changes:**

1. How were the thresholds (confidence ceiling τ_conf, closure τ_close, stability window κ) selected? Can you provide a sensitivity analysis and guidelines for choosing them in new domains?

2. How robust are the structured-field extractions (h_t, c_t, source groups, contradiction status) to prompt or parser variations? Do you have inter-parser or inter-prompt agreement statistics?

3. Can you provide aggregate metrics beyond HCR—e.g., contradiction integration rate, source-independence at stop, and premature-closure frequency—so that readers can see which components drive stability changes?

4. In the main HoVer comparison, can you report paired statistical tests for HCR differences (as in Table 13 for external slices) and include multiple random seeds to assess variance?

5. Would a cost-normalized comparison (equal token/latency budgets) change the stability vs. accuracy conclusions relative to forced extra search?

6. If possible, please provide a clearer diagram of the methodology and the results.